# Effectiveness of SARS-CoV-2 Decontamination and Containment in a COVID-19 ICU

**DOI:** 10.3390/ijerph18052479

**Published:** 2021-03-03

**Authors:** Zarina Brune, Cyrus E. Kuschner, Joseph Mootz, Karina W. Davidson, Robert C. F. Pena, Mustafa H. Ghanem, Austin Fischer, Michael Gitman, Lewis Teperman, Christopher Mason, Lance B. Becker

**Affiliations:** 1Donald and Barbara Zucker School of Medicine at Hofstra/Northwell, Northwell Health, Hempstead, NY 11549, USA; zbrune@northwell.edu (Z.B.); ckuschner1@northwell.edu (C.E.K.); jmootz@northwell.edu (J.M.); kdavidson2@northwell.edu (K.W.D.); rpena2@pride.hofstra.edu (R.C.F.P.); mghanem@northwell.edu (M.H.G.); afischer1@northwell.edu (A.F.); mgitman@northwell.edu (M.G.); lteperman@northwell.edu (L.T.); 2North Shore University Hospital/Long Island Jewish Emergency Medical Department, Northwell Health, Manhasset, NY 11030, USA; 3Institute of Health Innovations and Outcomes Research, Feinstein Institutes for Medical Research, Northwell Health, Manhasset, NY 11030, USA; 4Department of Physiology and Biophysics, Weill Cornell Medicine, New York, NY 10065, USA; 5The HRH Prince Alwaleed Bin Talal Bin Abdulaziz Alsaud Institute for Computational Biomedicine, Weill Cornell Medicine, New York, NY 10021, USA

**Keywords:** environmental sampling, SARS-CoV-2, COVID-19, decontamination, qPCR

## Abstract

**Background:** Health care systems in the United States are continuously expanding and contracting spaces to treat patients with coronavirus disease 2019 (COVID-19) in intensive care units (ICUs). As a result, hospitals must effectively decontaminate and contain severe acute respiratory syndrome coronavirus 2 (SARS-CoV-2) in constructed and deconstructed ICUs that care for patients with COVID-19. We assessed decontamination of a COVID-19 ICU and examined the containment efficacy of combined contact and droplet precautions in creating and maintaining a SARS-CoV-2–negative ICU “antechamber”. **Methods:** To examine the efficacy of chemical decontamination, we used high-density, semi-quantitative environmental sampling to detect SARS-CoV-2 on surfaces in a COVID-19 ICU and COVID-19 ICU antechamber. Quantitative real-time polymerase chain reaction was used to measure viral RNA on surfaces. Viral location mapping revealed the distribution of viral RNA in the COVID-19 ICU and COVID-19 ICU antechamber. Results were further assessed using loop-mediated isothermal amplification. **Results:** We collected 224 surface samples pre-decontamination and 193 samples post-decontamination from a COVID-19 ICU and adjoining COVID-19 ICU antechamber. We found that 46% of antechamber objects were positive for SARS-CoV-2 pre-decontamination despite the construction of a swinging door barrier system, implementation of contact precautions, and installation of high-efficiency particulate air filters. The object positivity rate reduced to 32.1% and viral particle rate reduced by 95.4% following decontamination. Matched items had an average of 432.2 ± 2729 viral copies/cm^2^ pre-decontamination and 19.2 ± 118 viral copies/cm^2^ post-decontamination, demonstrating significantly reduced viral surface distribution (*p* < 0.0001). **Conclusions:** Environmental sampling is an effective method for evaluating decontamination protocols and validating measures used to contain SARS-CoV-2 viral particles. While chemical decontamination effectively removes detectable viral RNA from surfaces, our approach to droplet/contact containment with an antechamber was not highly effective. These data suggest that hospitals should plan for the potential of aerosolized virions when creating strategies to contain SARS-CoV-2.

## 1. Introduction

The emergence of severe acute respiratory syndrome coronavirus 2 (SARS-CoV-2) has had a major impact on economies and lives across the globe. Almost 30 million people have been confirmed to have coronavirus disease 2019 (COVID-19), and nearly 2.5 million people have died from the disease globally [1].

Initial surges of SARS-CoV-2 infections required hospitals to rapidly expand intensive care units (ICUs) dedicated to caring for patients with COVID-19, hereafter referred to as “COVID-19 ICUs”. These expansions have ranged from adapting pre-existing units into COVID-19 ICUs to constructing de novo COVID-19 treatment centers. As some states continue to battle surges in COVID-19, others have either maintained or seen a decline in cases, allowing them to reconvert adapted COVID-19 ICUs to their original function. This essential process allows hospitals to return to pre-pandemic levels of patient care and economic stability, as there was a drastic decrease in both hospital admissions and elective procedures during the initial COVID-19 surge [2,3].

As these facilities expand and contract, hospitals must maintain treatment areas free of SARS-CoV-2. However, efforts to inhibit spread of SARS-CoV-2 viral particles across hospital spaces have not been entirely successful [4,5]. Infected patients can spread SARS-CoV-2 and contaminate their surroundings mainly through respiratory droplet dispersion [6,7], although fecal shedding [8] and aerosolization [9,10] have also been reported. In COVID-19 ICUs, the risk of environmental contamination is exacerbated by commonly used aerosolizing practices and procedures, including endotracheal intubation, cardiopulmonary resuscitation, ventilation, and tracheotomies [11,12,13,14]. In addition, SARS-CoV-2 viral particles can persist on surfaces for up to 9 days with fomite-dependent half-lives ranging from 2.3–17.9 h on stainless steel, plastic, and nitriles—all surfaces commonly found in hospitals [15,16]. The potentially high infectivity of COVID-19 has generated uncertainty regarding how to manage patients diagnosed with COVID-19 while preventing in-hospital and intra-departmental dissemination of SARS-CoV-2 viral particles. Thus, hospitals must demonstrate that their stringent viral containment and decontamination strategies create facilities free from SARS-CoV-2.

At Northwell Health, the largest integrated health care system in New York state, we rapidly constructed COVID-19 ICUs during the initial wave of infections that made New York City an epicenter in the pandemic. Within a few weeks, the North Shore University Hospital, part of the Northwell Health System, expanded from 85 adult ICU beds to 166 ICU beds. Of these, 23 were located in the previously designated Cardiac Short Stay Unit (CSSU). To minimize droplet and contact spread from patient-care zones, we constructed a COVID-19 ICU “antechamber” that served as an entrance and exit for the COVID-19 ICU. This antechamber was created to maintain a location free of SARS-CoV-2 where providers treating critically ill patients in the COVID-19 ICU could safely don and doff personal protective equipment, perform electronic medical charting, and consume food and beverage. These antechambers contained contact precautions, barrier precautions, and high-efficiency particulate air (HEPA) filters to prevent SARS-CoV-2 dissemination.

In this study, we aimed to validate decontamination of the COVID-19 ICU and associated antechamber per World Health Organization (WHO) guidelines by measuring viral copies of SARS-CoV-2 in environmental samples collected from both areas of the former CSSU [17]. Additionally, we examined whether the antechamber with droplet and contact containment strategies effectively created a SARS-CoV-2–negative zone. While there remains no current reproducible gold-standard for assessing environmental infectivity, the assessment presented here provides valuable insight on SARS-CoV-2 virion spread as well as barrier and decontamination efficacy.

## 2. Materials and Methods

### 2.1. Description of the COVID-19 ICU

The North Shore University Hospital CSSU was converted to a COVID-19 ICU to treat patients diagnosed with COVID-19. The original unit architecture included 3 provider workstations, 2 bathrooms, and space for 26 beds. The adapted COVID-19 ICU was an open, non-negative pressure unit that contained 1 provider workstation, 2 bathrooms, and 21 beds. The adjoining COVID-19 ICU antechamber contained 2 workstations protected by a contiguous metal barrier with plexiglass dividers, HEPA air-filters, and 2 swinging doors that allowed entrance to patient care. At the time of pre-decontamination sampling, 14 patients with COVID-19 were being actively treated in the COVID-19 ICU, in comparison to 0 patients present immediately post-decontamination during repeat sampling.

### 2.2. Decontamination Procedure

Standard chemical decontamination was performed by trained hospital cleaning staff and included the use of hydrogen peroxide Oxivir wipes (Diversey Global, Fort Mill, SC, USA) for general surfaces, ammonium chloride–based solutions (Virex II 256) for flooring, and ammonium chloride–based wipes (Chlorox) for bathroom surfaces.

### 2.3. Sample Processing

Pre-decontamination sampling occurred on 8 May 2020. Post-decontamination sampling occurred on 20 May 2020, 1 day after decontamination. Before sampling, polyester-tipped swabs (Isohelix) were pre-soaked in DNA/RNA Shield (Zymo Research, Irvine, CA, USA) for 1 min. These swabs were used to sample surfaces manually using consistent pressure for a timed 2-min period and then stored in prefilled vials containing 400 µL DNA/RNA Shield at 4 °C. Samples underwent viral RNA isolation and purification using the QIAamp Viral RNA Isolation Kit (Qiagen, Germantown, MD, USA).

### 2.4. Viral Load Analysis

Viral loads of samples were assessed with 1-step semi-quantitative reverse transcription polymerase chain reaction (qRT-PCR) using TaqPath 1-Step qRT-PCR Master Mix (Thermo Fisher Scientific, Waltham, MA, USA) on the LightCycler^®^ 480 System (Roche Diagnostics, Indianapolis, IN, USA). Samples were run in triplicate for each of the N1 and N2 primers and probes (Integrated DNA Technologies, Coralville, IA, USA) recommended by the Centers for Disease Control and Prevention. Quantification of viral RNA copies was accomplished using averaged N1 Ct values generated from a standard curve of known serial dilutions of SARS-CoV-2 N-gene-containing plasmid (2019-nCoV_N_Positive Control).

To calculate viral surface density, copy number was divided by the collection surface area. A minimum of 4 Ct values less than 38 was required for a sample to be considered positive.

### 2.5. Loop-Mediated Isothermal Amplification Assay

Loop-mediated isothermal amplification (LAMP) assays were performed using WarmStart^®^ LAMP Kit (DNA and RNA) (New England Biolabs, Ipswich, MA, USA), according to the manufacturer’s protocol. N-gene LAMP primer sequences were kindly provided by Nathan Tanner (New England Biolabs, Ipswich, MA, USA). Primers were ordered from Integrated DNA Technologies, with high-performance liquid chromatography purification specified for the forward and backward inner primer oligos. Reactions were supplemented with guanidine chloride to a final concentration of 40 mM [18]. Fluorescence was recorded using the LightCycler^®^ 480 System (Roche Diagnostics, Indianapolis, IN, USA).

### 2.6. Statistical Analysis

Viral load distribution pre- and post-decontamination was assessed using paired *t*-tests and Mann–Whitney U test. LAMP correlation was determined using linear regression. A Wilcoxon signed-rank test compared the efficacy of decontamination on matched samples.

## 3. Results

On 8 May 2020, we collected 224 samples from the COVID-19 ICU while it cared for patients with COVID-19 (pre-decontamination). On 20 May 2020, we collected another 193 samples after decontamination protocols were enacted to return the COVID-19 ICU to a non-COVID-19 CSSU (post-decontamination). Samples were collected from pre-specified objects within the COVID-19 ICU and COVID-19 ICU antechamber, including patient treatment areas, workstations, and bathrooms (Table 1). When possible, the same objects in each location were evaluated for viral surface distribution pre- and post-decontamination.

Implementation of standard chemical decontamination techniques per WHO recommendations significantly reduced viral loads throughout the CSSU (Figure 1).

We found that 57.6% (129/224) of samples carried quantifiable virus pre-decontamination compared to 23.3% (45/193) post-decontamination. More than 50% of objects near patients with COVID-19 had detectable levels of SARS-CoV-2 pre-decontamination that were reduced post-decontamination (Table 1). In the antechamber, 46% of sampled objects tested positive for SARS-CoV-2 viral RNA pre-decontamination, which reduced to 32.1% post-decontamination (Table 1). Object positivity-negativity was determined via the cutoff of a minimum of 4 Ct values less than 38 for consideration as a “positive object”. Significance of changes of overall object positivity in each treatment area were then determined using the Mann–Whitney U Test (*p* < 0.05). Calculated *p* values were as follows: significant differences were found in the treatment area (*p* < 0.00001) and workstation-treatment area (*p* < 0.00914) with no significant difference in the workstation-antechamber (*p* < 0.1814) and bathroom-treatment area (*p* < 0.3557).

Pre-decontamination objects carried viral loads between 10 and 100 viral copies/cm^2^, whereas post-decontamination objects predominately carried loads between 0 and 1 viral copies/cm^2^ (Figure 2A). In addition, when matched items were assessed, the viral surface distribution pre-decontamination (mean: 432.2 ± 2729 viral copies/cm^2^, interquartile range [IQR] 19.8) was significantly reduced by 95.6% post-decontamination (mean: 19.2 ± 118 viral copies/cm^2^, IQR 1.95) (*p* < 0.0001) (Figure 2B).

Interestingly, there was no significant difference between viral loads in the antechamber (3.79 ± 4.2 viral copies/cm^2^) and treatment areas (9.06 ± 14.61 viral copies/cm^2^) pre-decontamination (*p* = 0.346). Most items carried viral loads between 1 and 10 viral copies/cm^2^ both pre- and post-decontamination (Table 2). The highest viral loads were found on doorknobs to enter antechamber workstations. Furthermore, the plexiglass barricade separating providers from treatment areas tested positive for SARS-CoV-2 RNA on both the protected-provider side and the patient-treatment side. Post-decontamination, the number of items contaminated in all workstations was reduced. While most surfaces that were positive post-decontamination had viral loads of 0 to 10 viral copies/cm^2^, doorknobs and keyboards remained positive with 10 to 100 viral copies/cm^2^. This latter finding illustrates the critical importance of identifying potential failures in decontamination with special attention paid to objects known to have increased physical contact with viral carriers (e.g., doorknobs, keyboards).

Bathrooms were negative for any viral positivity pre-decontamination. Post-decontamination, toilet flush handles tested positive (Table 1).

We used semi-quantitative LAMP to validate qRT-PCR viral quantification for 160 samples (80 pre-decontamination and 80 post-decontamination). The qRT-PCR results positively correlated with LAMP results (regression Y = 0.1807X + 29.68) (Figure 3).

## 4. Discussion

In this study, we provide the first environmental assessment of the effectiveness of SARS-CoV-2 decontamination in a COVID-19 ICU and COVID-19 ICU antechamber. From our data, we generated a before-and-after heatmap demonstrating the quantity and location of SARS-CoV-2 viral RNA on high-risk fomites pre- and post-decontamination (Figure 1). These data illustrate that standard chemical decontamination per WHO guidelines effectively removes SARS-CoV-2 viral RNA from surfaces in a hospital environment [17]. It also indicates that a COVID-19 ICU antechamber was ultimately unsuccessful at containing contamination, implicating that droplet precautions, HEPA air filters, and contact safeguards may not sufficiently mitigate the spread of SARS-CoV-2.

Initially, methods of SARS-CoV-2 transmission were hotly debated. While early studies predominantly focused on the potential for droplet transmission, recent literature supports aerosol transmission [6,19,20]. The antechamber sampled in this study had interventions in place to limit droplet dispersion. However, our results demonstrate no significant differences in viral distribution between the antechamber and the COVID-19 ICU. We found that the number of viral particles were nearly equal on both sides of plexiglass dividers, which suggests failure of the antechamber containment setup and a potential spread of viral particles via aerosolization rather than sole air droplet transmission. This finding supports that hospitals need to at least consider utilizing aerosol-based strategies in addition to droplet- and contact-based strategies for containment of SARS-CoV-2 particles.

The fomites we detected and characterized are consistent with previous studies that collected environmental samples. Nebraska Medical Center implemented intermittent sampling of medical floors and isolation rooms. They identified several high-risk fomites, including surface samples, such as room surfaces and toilets, as well as high- and low-volume air samples [19]. In addition, a hospital in Wuhan, China used viral mapping to demonstrate contamination of surfaces in the ICU [21]. Our heatmap demonstrates the efficacy of standard chemical techniques in decontaminating surfaces exposed to SARS-CoV-2 in a hospital setting. The literature has also described alternative methods for SARS-CoV-2 elimination, including high-intensity ultraviolet light, ozone gas, and other decontamination agents for surface, droplet-borne, and aerosolized SARS-CoV-2 [22,23,24,25]. However, research is only now beginning to emerge with regard to their application as a widespread, standardized replacement for current chemical methods and should continue to be investigated [26].

As society attempts to reopen amid waves of viral resurgence, we need techniques capable of widespread viral monitoring and validating decontamination. We further supported the validity of our qRT-PCR results using fluorescent-based, semi-quantitative LAMP. Our previous work showed that LAMP has similar sensitivity to qRT-PCR and reduces the processing time to less than 30 min [27,28,29]. LAMP results can also be visualized using colorimetric assays, including fluorescent dyes that allow for semi-quantitative analysis of viral load in each sample. Our data support that LAMP is a potential alternative to qRT-PCR for rapidly processing and detecting SARS-CoV-2 [30,31].

This study had several limitations. While we present detailed data on viral copy numbers and heat maps of viral distribution, we did not conduct infectivity assays. So far, there has been no direct correlation shown between Ct values from SARS-CoV-2 environmental surface or air samples and sample infectivity. However, recent studies used patient samples to examine the ability of Ct values to predict infectivity. These studies showed an odds ratio for positive viral culture of 0.64 for every 1 unit increase in Ct value, with no infectivity detected in samples with a Ct value greater than 24 [32]. Other studies using patient samples demonstrated correlations between infectivity and duration of infection but no significant correlation with Ct value [33]. Future work analyzing environmental SARS-CoV-2 samples should perform similar studies to assess relationships between viral genomic copy number and infectivity. In addition, although surfaces were sampled and measured following standard protocols to minimize error, some variability in surface collection occurred due to inherent irregularity of surfaces. Air sampling for SARS-CoV-2 was performed; however, the results were inconclusive for viral positivity and thus not included in this study.

## 5. Conclusions

In summary, we established the utility of environmental sampling in objectively assessing the WHO-recommended decontamination procedures used by hospitals to disinfect COVID-19 ICUs and prevent the dissemination of SARS-CoV-2. It is increasingly apparent that the three arms of SARS-CoV-2 dissemination—direct-contact transmission, droplet-containing virions, and aerosolized-viral particles—must all be targeted to effectively control the continued spread of SARS-CoV-2 infection. Future studies should compare the efficacy of various combinations of interventions in mitigating the dispersion of SARS-CoV-2.

## Figures and Tables

**Figure 1 ijerph-18-02479-f001:**
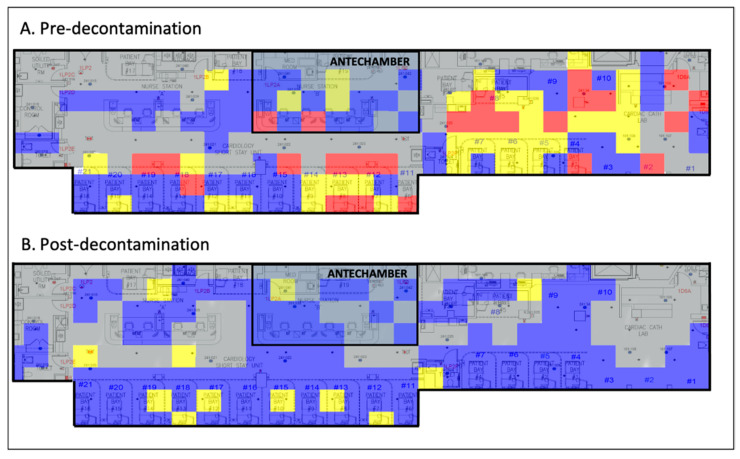
Viral heat map illustrating the efficacy of chemical decontamination of a COVID-19 ICU. Grey, no sample taken; blue, 0–1 viral copies/cm^2^; yellow, 1–10 viral copies/cm^2^; red, >100 viral copies/cm^2^. COVID-19, coronavirus disease 2019; ICU, intensive care unit.

**Figure 2 ijerph-18-02479-f002:**
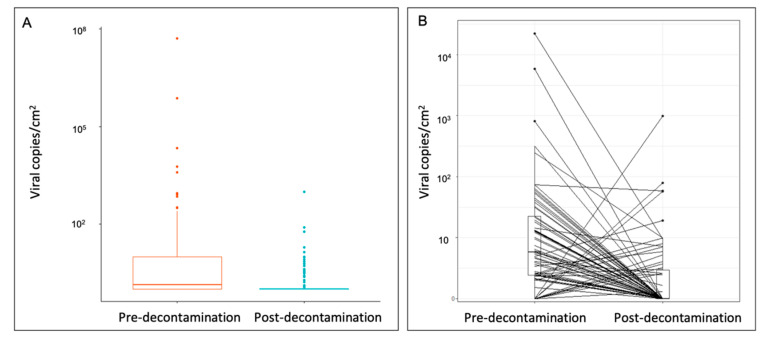
Quantification of SARS-CoV-2 surface distribution on (**A**) all sampled objects and (**B**) matched sampled objects pre- and post-decontamination. Wilcoxon signed-rank test on matched samples (*p* < 0.0001). SARS-CoV-2, severe acute respiratory syndrome coronavirus 2.

**Figure 3 ijerph-18-02479-f003:**
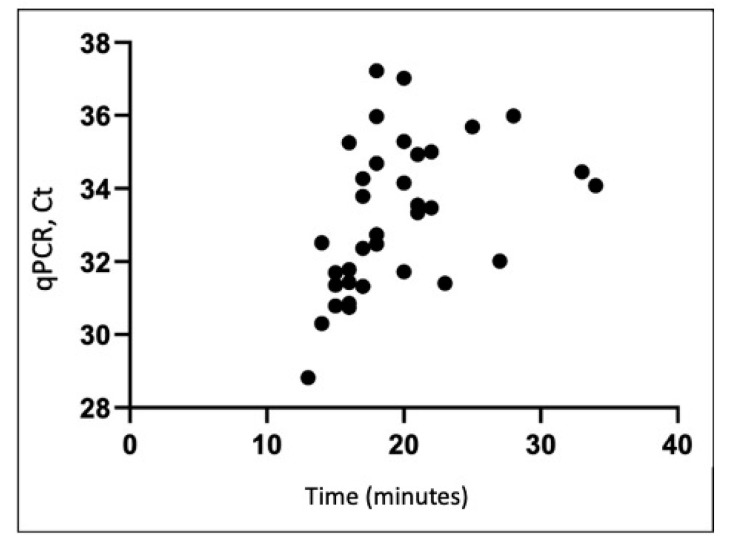
Comparison of qRT-PCR Ct values and LAMP detection of SARS-CoV-2 RNA in collected environmental samples (*R*^2^ = 0.19). LAMP, loop-mediated isothermal amplification; qRT-PCR, quantitative real-time polymerase chain reaction; SARS-CoV-2, severe acute respiratory syndrome coronavirus 2.

**Table 1 ijerph-18-02479-t001:** Viral Distribution of Objects Positive for SARS-CoV-2 Pre- and Post-Decontamination.

	Pre-Decontamination, No. (%) (n = 224)	Post-Decontamination, No. (%) (n = 193)
Treatment Area	n = 168	n = 140
Bedrails	15/20 (75%)	2/14 (14%)
Cardiac monitor	9/19 (47%)	3/19 (16%)
Ceiling	1/4 (25%)	0/4 (0%)
Curtain	0/5 (0%)	NA
Doorknob	1/1 (100%)	NA
Floor	21/26 (81%)	10/23 (44%)
IV pole	10/16 (63%)	3/17 (18%)
Keyboard	5/8 (63%)	0/9 (0%)
Kitchen	3/5 (60%)	0/5 (0%)
Medication dispenser	13/21 (62%)	NA
Monitor	5/9 (56%)	0/10 (0%)
Patient table	4/6 (60%)	3/15 (20%)
Protective windows	3/4 (75%)	NA
Purell dispenser	0/1 (0%)	0/2 (0%)
Sharps storage	1/2 (50%)	0/6 (0%)
Supply closet	1/2 (50%)	0/2 (0%)
Ventilator screen	3/4 (75%)	NA
Wall	8/15 (53%)	3/14 (21%)
Workstation Antechamber	n = 35	n = 28
Doorknob	1/2 (50%)	2/2 (100%)
Floor	6/6 (100%)	2/4 (50%)
Keyboard/Mouse	3/9 (33%)	2/6 (33%)
Monitor	4/9 (44%)	0/6 (0%)
Purell dispenser	0/2 (0%)	0/1 (0%)
Surfaces	2/7 (29%)	2/6 (33%)
Wall	NA	1/3 (33%)
Workstation—Treatment Area	n = 13	n = 17
Doorknob	0/1 (0%)	0/1 (0%)
Floor	1/1 (100%)	1/2 (50%)
Keyboard/Mouse	2/4 (50%)	0/4 (0%)
Monitor	3/4 (75%)	1/5 (20%)
Pixus screen	1/1 (100%)	0/2 (0%)
Workstation surfaces	2/2 (100%)	1/2 (50%)
Workstation walls	NA	0/1 (0%)
Bathroom—Treatment area	n = 8	n = 8
Ceiling	0/2 (0%)	0/2 (0%)
Doorknob	0/2 (0%)	0/2 (0%)
Toilet flush handle	0/2 (0%)	1/2 (50%)
Wall	0/2 (0%)	0/2 (0%)

Abbreviations: NA, not applicable.

**Table 2 ijerph-18-02479-t002:** Quantitative Distribution of SARS-CoV-2–Positive Fomites Pre- and Post-Chemical Decontamination.

Viral Copies	Pre-Decontamination, No. (%)	Post-Decontamination, No. (%)
10 viral copies/cm^2^	71 (55%)	38 (84%)
10–100 viral copies/cm^2^	42 (33%)	6 (13%)
>100 viral copies/cm^2^	16 (12%)	1 (2%)

## Data Availability

The data that support the findings of this study are available on request from lance.becker@northwell.edu.

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
