# Peer review of "Effectiveness of SARS-CoV-2 Decontamination and Containment in a COVID-19 ICU"

_ijerph, 2021, doi:10.3390/ijerph18052479_

Round 1

Reviewer 1 Report

The present paper is the first environmental assessment (as the authors have claimed) and a valuable case study that provides useful insights for COVID-19 control in medical institutions. In particular, it is very important to point out that antechamber is not effective when it comes down to it and that previous measures such as droplet precautions and HEPA filters have their limitations. In this respect, the paper deserves to be accepted, however the reviewer would like to request the following minor revisions beforehand.

1. In the Decontamination procedure in the Materials and methods, there need reference(s) to claim as a standard chemical decontamination. Additionally, the potential applicability of other methods than chemical decontamination, such as ozone gas decontamination, should be mentioned in more detail in the Introduction and/or Discussion. This is necessary for non-specialist readers.

2. In Table 1, for each treatment area, significant differences in pre/post-decontamination should be described with p-values. It would also be helpful to discuss whether the differences in the treatment area are simply due to under-sampling, or whether there is an essential reason for some areas, such as doorknobs or keyboards.

3. In the matched sampled objects in Figure 2, there are cases of an increase in post-decontamination: how can this be interpreted?

Author Response

Dear Reviewer 1,

Thank you so much for taking the time to read and evaluate this manuscript. I greatly appreciate your feedback and made the suggested changes. Responses to each of your points are listed below.

“The present paper is the first environmental assessment (as the authors have claimed) and a valuable case study that provides useful insights for COVID-19 control in medical institutions. In particular, it is very important to point out that antechamber is not effective when it comes down to it and that previous measures such as droplet precautions and HEPA filters have their limitations. In this respect, the paper deserves to be accepted, however the reviewer would like to request the following minor revisions beforehand.”

  1. In the Decontamination procedure in the Materials and methods, there need reference(s) to claim as a standard chemical decontamination. Additionally, the potential applicability of other methods than chemical decontamination, such as ozone gas decontamination, should be mentioned in more detail in the Introduction and/or Discussion. This is necessary for non-specialist readers.

The WHO guidelines are cited as standard decontamination methods. We also included further references to alternative methods of decontamination briefly in the discussion.

  1. In Table 1, for each treatment area, significant differences in pre/post-decontamination should be described with p-values. It would also be helpful to discuss whether the differences in the treatment area are simply due to under-sampling, or whether there is an essential reason for some areas, such as doorknobs or keyboards.

We included p-values as requested. We also provided a potential explanation for the differences in the treatment area as suggested.

  1. In the matched sampled objects in Figure 2, there are cases of an increase in post-decontamination: how can this be interpreted?

We provided a theory – there could have been a failure of decontamination. This is supported by the “high frequency touch rate” nature of the object, increasing the likelihood that the viral load increases over time without adequate cleaning.

Reviewer 2 Report

The objective of the current study was the investigate utility of environmental sampling assessing the decontamination procedures used by hospitals to disinfect COVID-19 ICUs and to prevent the dissemination of SARS-CoV-2. In my opinion, some comments should be addressed:

Line 43: “Results were validated using loop-mediated isothermal amplification”

Authors cannot validate results of real-time PCR examination by using LAMP technique. Validation is a complicated process that require comparison of new method (during validation) to reference method. In SARS-CoV-2 examination reference method is real-time PCR or virus culture. I strongly suggest to change this sentence

Introduction:

Line 62: “….and nearly 1 million people have died from the disease” – it is not true now. Please use recent datas.

Line 75: “Infected patients can spread SARS-CoV-2 through a combination of respiratory droplet dispersion, fecal shedding, and aerosolization, contaminating the environment where they are treated.” According to present knowledge, there is only one infectious route – droplet route, so authors, in my opinion, cannot write that SARS-CoV-2 is spread through fecal shedding or aerosolization. I believe that it is possible, but it has not been proved.

Line 80: “… COVID-19 can persist on surfaces for up to 9 days”

  • COVID-19 is a disease. On surfaces can persist virus.
  • persistence of SARS-CoV-2 on surfaces is depend on various factors, i.e. environmental condition, type of surface.
  • SARS-CoV-2 can persist even more than 9 days. Please find more present information about this topic.

Materials and methods

Line 114: “At the time of pre-decontamination sampling, 14 patients with COVID-19 were being actively treated in the COVID-19 ICU”.

Were there any patients at the time of post-decontamination sampling? If yes, were there the same patients or patients in the same health condition as during pre-decontamination sampling? Patients in various health condition or absence of patients in the ICU (and I suspect that there were no patients in the clinic if post-sampling was done before return the ICU do CSSU) can generate different level of viral compounds to the environment, and it can impact on results.

Line 118: Decontamination was done by who? Professional staff after training or researchers?

Results

General:

  • Please explain, if obtained result was 1 copy/cm2, it was positive results or negative? Refer the cut-off for positive and negative result of real-time PCR - I do not understand some sentences, i.e. “While most surfaces that were positive post-decontamination had viral loads of 0 to 10 viral copies/cm2” – it means that 0 is a positive result?
  • Please explain, why did you use different scales on figures 2a and 2b? If I correctly understand, on both figures are the same results.

Line 274: “We found that the number of viral particles were nearly equal on both sides of plexiglass dividers, which suggests aerosolized viral particles rather than droplet or contact spread of the viral particles.” I think that it is too brave conclusion – it can suggests also that this antechamber doesn’t work.

Line 286: “We confirmed our qRT-PCR…” Authors can not confirm results of PCR by using LAMP. Please, change the sentence.

Author Response

Dear Reviewer 2,

Thank you so much for taking the time to read and evaluate this manuscript. I greatly appreciate your valuable suggestions and have addressed each of them below.

“The objective of the current study was the investigate utility of environmental sampling assessing the decontamination procedures used by hospitals to disinfect COVID-19 ICUs and to prevent the dissemination of SARS-CoV-2. In my opinion, some comments should be addressed:”

Line 43: “Results were validated using loop-mediated isothermal amplification”

Authors cannot validate results of real-time PCR examination by using LAMP technique. Validation is a complicated process that require comparison of new method (during validation) to reference method. In SARS-CoV-2 examination reference method is real-time PCR or virus culture. I strongly suggest to change this sentence

Reply: This sentence has been changed to show LAMP further supported the results, rather than “validated”.

Introduction:

Line 62: “….and nearly 1 million people have died from the disease” – it is not true now. Please use recent datas.

Reply: We have updated this data.

Line 75: “Infected patients can spread SARS-CoV-2 through a combination of respiratory droplet dispersion, fecal shedding, and aerosolization, contaminating the environment where they are treated.” According to present knowledge, there is only one infectious route – droplet route, so authors, in my opinion, cannot write that SARS-CoV-2 is spread through fecal shedding or aerosolization. I believe that it is possible, but it has not been proved.

Reply: We changed the phrasing to emphasize the infectious route as being mainly through droplet, with studies demonstrating dispersion via these other methods.

Line 80: “… COVID-19 can persist on surfaces for up to 9 days”

  • COVID-19 is a disease. On surfaces can persist virus.
  • persistence of SARS-CoV-2 on surfaces is depend on various factors, i.e. environmental condition, type of surface.
  • SARS-CoV-2 can persist even more than 9 days. Please find more present information about this topic.

Reply: We have changed COVID-19 to SARS-CoV-2. We also elaborated upon viral half-life on hospital specific fomites per reviewer request and included additional sources for support.

Materials and methods

Line 114: “At the time of pre-decontamination sampling, 14 patients with COVID-19 were being actively treated in the COVID-19 ICU”.

Were there any patients at the time of post-decontamination sampling? If yes, were there the same patients or patients in the same health condition as during pre-decontamination sampling? Patients in various health condition or absence of patients in the ICU (and I suspect that there were no patients in the clinic if post-sampling was done before return the ICU do CSSU) can generate different level of viral compounds to the environment, and it can impact on results.

Reply: There were no patients following sampling – this was clarified in the manuscript.

Line 118: Decontamination was done by who? Professional staff after training or researchers?

Reply: We clarified that decontamination was performed by trained hospital cleaning staff.

Results

General:

  • Please explain, if obtained result was 1 copy/cm2, it was positive results or negative? Refer the cut-off for positive and negative result of real-time PCR - I do not understand some sentences, i.e. “While most surfaces that were positive post-decontamination had viral loads of 0 to 10 viral copies/cm2” – it means that 0 is a positive result?

Reply: Although this point was initially discusses in the methods section, we further clarified in the discussion what was considered a positive viral test results.

  • Please explain, why did you use different scales on figures 2a and 2b? If I correctly understand, on both figures are the same results.

Reply: The different scale was used to help better visualize the results graphically.

Line 274: “We found that the number of viral particles were nearly equal on both sides of plexiglass dividers, which suggests aerosolized viral particles rather than droplet or contact spread of the viral particles.” I think that it is too brave conclusion – it can suggests also that this antechamber doesn’t work.

Reply: We included in the discussion this very important point that you bring up – failure of the antechamber - in addition to the potential for aerosolization (sources included).

Line 286: “We confirmed our qRT-PCR…” Authors can not confirm results of PCR by using LAMP. Please, change the sentence.

Reply: This sentence has been changed as requested.

Round 2

Reviewer 2 Report

Dear Authors,

thank you for your reply and compliance my comments. I think that your manuscript can be published in the current form.

Kind regards